# Learning Object-Oriented Dynamics for Planning from Text

**Guiliang Liu**[1,3] , **Ashutosh Adhikari**[1,4] , **Amir-massoud Farahmand**[2,3], **Pascal Poupart**[1,3]
[1]University of Waterloo, [2]University of Toronto, [3]Vector Institute, [4]Microsoft
g233liu@uwaterloo.ca, aadhikari@microsoft.com,
farahmand@vectorinstitute.ai,ppoupart@uwaterloo.ca

## Abstract

The advancement of dynamics models enables model-based planning in complex environments. Dynamics models mostly study image-based games with fully observable states. Generalizing these models to Text-Based Games (TBGs), which often include partially observable states with noisy text observations, is challenging. In this work, we propose an Object-Oriented Text Dynamics (OOTD) model that enables planning algorithms to solve decision-making problems in text domains. OOTD predicts a memory graph that dynamically remembers the history of object observations and filters object-irrelevant information. To improve the robustness of dynamics, our OOTD model identifies the objects influenced by input actions and predicts beliefs of object states with independently parameterized transition layers. We develop variational objectives under the object-supervised and self-supervised settings to model the stochasticity of predicted dynamics. Empirical results show that our OOTD-based planner significantly outperforms model-free baselines in terms of sample efficiency and running scores.

## 1 Introduction

Planning algorithms typically leverage the environment dynamics to solve decision-making problems (Sutton & Barto, 2018). To plan in unknown environments, the agent must learn a dynamics model to predict future (belief) states and rewards by conditioning on an action. This dynamics model enables the implementation of intensive search for optimal actions, which can potentially increase both sample efficiency and cumulative rewards, compared to model-free methods (Hafner et al., 2019; Wang et al., 2019; Kaiser et al., 2020). Despite the promising performance, learning a dynamics model that can accurately generalize at test time is still challenging, especially when handling a high-dimensional state space for low-level features, e.g., pixels and text.

To facilitate dynamics learning in complex environments, Diuk et al. (2008) proposed an Object-Oriented Markov Decision Process (OO-MDP) that factorizes world states into object states. They showed that the agent can find optimal policies with a better sample efficiency by modeling the dynamics at the object level. Some following works (Finn et al., 2016; Goel et al., 2018; Zhu et al., 2018; 2020) extended OO-MDPs to image-based games. These methods typically assume *full* observability over game states and a *fixed* input size, which facilitates the use of object masks to decompose an image into different objects. On the other hand, in Text-Based Games (TBGs), the text observation, whose length runs from a few words to an entire paragraph, is a *partial* description of the current game state, and thus each observation provides information about a limited number of objects (in fact, only an average of $4.51\%$ of the candidate objects are mentioned in an observation from Textworld (Côté et al., 2018)). The dynamics model must remember the history of observed objects to predict accurately their states. Moreover, an observation typically contains lots of noisy patterns that record object-irrelevant information (e.g., the non-bolded text from $o_t$ in Figure 1). While previous works (Ammanabrolu & Riedl, 2019; Ammanabrolu & Hausknecht, 2020) designed rule-based heuristics to extract useful information, we expect that dynamically capturing object information from noisy observations helps to generalize to more environments.

In this work, we design an Object-Oriented Text Dynamics (OOTD ) model that integrates:

*1) Graph Representation for Objects.* OOTD predicts a memory graph from the input states. The memory graph, whose nodes correspond to the objects to be modelled, captures the information about these objects from the beginning of a game. By applying this graph as an *information bottleneck* for

dynamics prediction, OOTD filters the object-irrelevant information in the text inputs, which forms a tighter information plane and facilitates generalization to unseen games (Tishby & Zaslavsky, 2015).

*2) Independent Transition Layers.* Intuitively, an input action influences the states of a limited number of objects, for example, when the agent performs "*take the carrot from the counter*", only the states of "*carrot*" and "*counter*" should be updated. To capture which objects are influenced and keep others *invariant* to this update, OOTD learns an action-aware representation for each object with bi-directional attention and predicts the belief states of objects with independently parameterized transition layers. This independent mechanism enhances the robustness for dynamics prediction (Goyal et al., 2019) (also see the ablation study in Section 4.2).

During planning, we predict belief states without knowing rewards and observations. To learn a stochastic dynamics model, we propose object-supervised and self-supervised Evidence Lower Bound (ELBo) objectives for training our OOTD model. We evaluate how well the OOTD model supports planning by implementing planning algorithms (e.g., Dyna-Q (Kuvayev & Sutton, 1996) and Monte Carlo Tree Search (MCTS) (Kocsis & Szepesvári, 2006)) based on learned dynamics. Empirical results show that these planning algorithms significantly outperform other baselines in terms of game performance and sample efficiency. To support the design of our OOTD model, we include an ablation study that visualizes the object states and quantifies the dynamics prediction performance.

**Contributions:** 1) We introduce our approach to implementing model-based planning algorithms for solving decision-making problems in the text-domain. 2) We propose the OOTD model that learns to predict beliefs about object dynamics.

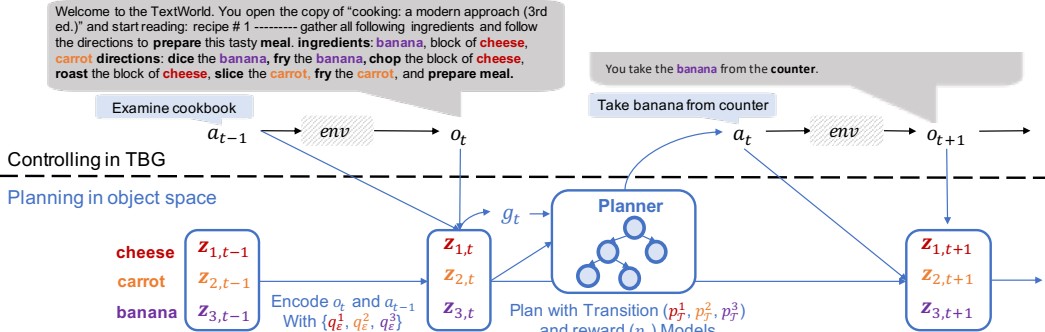

Figure 1: An example of planning from text. We show three objects (cheese, carrot and banana) in this example. When interacting with the environment ($env$), the agent encodes the information of observation $o_t$ into object states $z_t$ with encoders ($q_{\mathcal{E}}^1, q_{\mathcal{E}}^2, q_{\mathcal{E}}^3$). Given a goal $g_t$, the planner determines the action $a_t$ by searching with the reward ($p_r$) and transition models ($p_{\mathcal{T}}^1, p_{\mathcal{T}}^2, p_{\mathcal{T}}^3$).

## 2 OBJECT LEVEL PLANNING IN TEXT DOMAINS

In this work, the object-level dynamics are learned and evaluated in Text-Based Games (TBGs), for which we extend OO-MDPs (Diuk et al., 2008) to capture partial observability. We define a graph representation to remember object information and introduce the corresponding planning algorithms.

### 2.1 TEXT-BASED GAMES

Text-Based Games (TBGs) are complex, interactive simulations where an agent explores and plays a game by processing text observations and responding with text commands. To study the generalization ability of this policy, many recent works (Yin & May, 2019; Hausknecht et al., 2019; Ecoffet et al., 2019; Adolphs & Hofmann, 2020; Adhikari et al., 2020) proposed to learn a policy by training with a set of games and evaluating with games from a hold-out test set. The distributions of rewards over state-action pairs are different in training and testing games because TBGs assign rewards by conditioning on specific goals (that are predefined, but unknown). In this setting, TBGs have different intermediate goals in each game, but they commonly share the same ultimate goals, and the underlying causal dependencies leading to this ultimate goal are consistent. For example, to solve the 'First Text-World Problems' (Côté et al., 2018), an agent should always gather and process cooking ingredients (e.g., "*fry the potato*") according to a recipe that it discovers in the game, although the names and locations of the ingredients are different across different games. This property enables the generalization of knowledge learned from training games to solve testing games.

## 2.2 Object-Oriented Partially Observable Markov Decision Processes

Since an observation reveals only partial information about the game state, we formulate TBGs as partially observable Markov decision processes (POMDPs). In this work, we extend the POMDP framework to represent object-level dynamics (Diuk et al., 2008). The Object-Oriented (OO) POMDP is a tuple $\langle \mathcal{S}, \mathcal{O}, \mathcal{Z}, \Phi, \mathcal{G}, \mathcal{A}, \mathcal{R}, \mathcal{T}, \gamma \rangle$, where:

- $\mathcal{S}$ and $\mathcal{O}$ define the spaces of *low-level* states and observations from the TGB environments while $\mathcal{Z}$ and $\Phi$ are the spaces of *object-level* states and observations. In a TBG, $o \in \mathcal{O}$ is a text message from the game environment (e.g., "*Please fry the potato.*"), whereas $\phi \in \Phi$ records the specific objects and their relations in this message (e.g., a triplet like "*potato-need-fry*")). To model the object dynamics, the agent must extract $\phi$ from $o$ by distilling objects' information from text sentences. In this work, we model *object states* $z \in \mathcal{Z}$ (instead of $s \in \mathcal{S}$) and learn *latent representations* for $z$.

- $\mathcal{G}$ and $\mathcal{A}$ are the spaces for goals and actions. We study *choice-based games* (Yin & May, 2019), where candidate actions $A_t \in \mathcal{A}$ are available at time step $t$. We include a goal variable $g$ to mark different tasks in each game. Following the Universal MDP (UMDP) (Schaul et al., 2015), the agent initializes a goal at the beginning of a game and updates it when the task is finished.

- $\mathcal{R}$ and $\mathcal{T}$ define the spaces of reward and transition models. $\gamma$ is a discount factor. We assume the real dynamics models ($p_{\mathcal{T}}^*$ and $p_r^*$) are unknown, so we learn the object-oriented transition model and the reward model, i.e., $\forall k \in \{1, \ldots, K\}$, $p_{\mathcal{T}}^k(z_{k,t}|\boldsymbol{z}_{t-1}, a_{t-1}) \in \mathcal{T}$ and $p_r(r_t|\boldsymbol{z}_{t-1}, g_t) \in \mathcal{R}$.

In our OO-POMDP, transitions $p_{\mathcal{T}}$, observations $\phi$, and states $z$ are modelled for each object, whereas actions $a$ and rewards $r$ are defined for the entire environment. This generalizes to popular RL environments that accept an action and return a reward at every time step. Unlike (Wandzel et al., 2019) that represented object states with symbolic attributes, we develop a latent object representation that can generalize to the complex environment with high-dimensional inputs (e.g., text).

## 2.3 Graph Representation for Objects

We utilize an *object-relation-object* triplet $\phi$ to represent object information. At time step t, $\boldsymbol{\phi_t} = [\phi_{t,1}, ..., \phi_{t,M}]$ ($M$ is the number of observed triplets at $t$). Given a total of $K$ candidate objects and $C$ candidate relations, these triplets can be mapped to a knowledge graph $\Omega_t \in \{0,1\}^{C \times K \times K}$. Each entry $(c, i, j) \in \{0, 1\}$ indicates whether there is a relationship $r$ between the $i^{th}$ and $j^{th}$ objects. This knowledge graph forms a natural representation of object relations since object information in most observations $o_t$ corresponds to either entity attributes (e.g., "*potato-is-fried*") or to relational information about entities (e.g., "*potato-on-counter*" and "*bedroom-north_of-kitchen*"), which can be conveniently represented by triplets and the corresponding graph.

**Memory Graph.** We store the observations up to time step $t$ in a memory graph $h_t = \Omega_0 \oplus \Omega_1 \oplus \cdots \oplus \Omega_t$ that captures object information observed by the agent since the beginning of a game. Similar to previous works Adhikari et al. (2020); Ammanabrolu & Riedl (2019); Ammanabrolu & Hausknecht (2020), we summarize the object-level history with a latent memory graph $h_t$. To update the memory from $t-1$ to $t$, we learn a graph updater $\oplus$, and $h_t = h_{t-1} \oplus \Omega_t$. During updating, $\oplus$ needs to resolve some semantically contradictory triplets, for example "*player-at-kitchen*" and "*player-at-bedroom*" (because the player cannot simultaneously be in two different locations). Our OOTD model is trained to automatically emulate such an operator. Our transition model (Section 3.1) is trained to generate $h_t$ from $\boldsymbol{z}_t$, allowing the latent object states to capture object relations.

## 2.4 Model-Based Planning in Text-Based Games

We introduce model-based planning in TBGs. Based on the OO-POMDP (Section 2.2), at each time step $t$, we define latent states $\boldsymbol{z}_t = [z_{1,t}, \ldots, z_{K,t}]$ for a total of K objects, text observations $o_t$, action commands $a_t$, and scalar rewards $r_t$, that follow the stochastic dynamics in Table 1. The details of these dynamics models are introduced in Section 3.

Table 1: Object dynamics for planning, where the transition models and observation encoders are independently parameterized for a total of $K$ objects.

| | |
|---|---|
| Transition models | $\left[p_{\mathcal{T}}^k(z_{k,t+1}|a_t, \boldsymbol{z}_t)\right]_{k=1}^K$ |
| Observation encoders | $\left[q_{\mathcal{E}}^k(z_{k,t+1}|o_{t+1}, a_t, \boldsymbol{z}_t)\right]_{k=1}^K$ |
| Reward model | $p_r(r_t|\boldsymbol{z}_t, g_t)$ |
| Graph & Obs. decoder | $p_\Omega(h_t|\boldsymbol{z}_t), p_o(o_t|\boldsymbol{z}_t)$ |

Based on these dynamics, we implement planning algorithms to select an action from candidate commands $A_t$ for maximizing the expected sum of rewards $\mathbb{E}_\pi(\sum_{t=0}^T \gamma^t r_t)$, as shown in Figure 1. In this work, we study *choice-based games* (Yin & May, 2019), where the candidate commands (or actions) $A_t$ are available and the planner determines the action $a_t \in A_t$ to be performed. The planning algorithms include Dyna-Q, Monte-Carlo Tree Search (MCTS) and their combinations:

**Dyna-Q** (Kuvayev & Sutton, 1996) incorporates dynamics models and Q-learning. Dyna-Q interactively 1) updates the dynamics model with observed transitions and 2) trains the Q network to minimize the Temporal Difference (TD) loss (Equation 1) based on both *observed* transitions from the environment and *predicted* transitions from the dynamics models. Compared to the model-free Deep Q-Network (DQN), Dyna-Q is more sample efficient: by expanding the replay buffer with dynamics models, Dyna-Q converges faster with the same number of interactions from the environment.

$$\mathcal{L}_{TD} = \mathbb{E}\big[\big(r_t + \gamma \max_{a \in A_t} Q(\boldsymbol{z}_t, a, g_t) - Q(\boldsymbol{z}_{t-1}, a_{t-1}, g_{t-1})\big)^2\big] \tag{1}$$

**MCTS** (Kocsis & Szepesvári, 2006) is a heuristic search algorithm that builds and updates a search tree based on environment dynamics. By performing Monte-Carlo simulations organized in a tree structure, MCTS does an efficient search in environments with large action spaces (Silver et al., 2018). MCTS iteratively runs multiple playouts. At the $i^{th}$ playout, we implement 1) *Selection:* Traverse the tree from the root node to a leaf node (corresponding to object state $\boldsymbol{z}_\tau$) by selecting the action command $a_{i,t}$ to maximize the Upper Confidence Bound (UCB) (Couëtoux et al., 2011):

$$a_{i,t} = \arg\max_{a \in A_t} \Big[ Q_i(\boldsymbol{z}_t, a, g) + c_{puct} \frac{\sqrt{\log(i-1)}}{\zeta_{i-1}(\boldsymbol{z}_t, a, g) + 1} \Big] \tag{2}$$

where $c_{puct}$ controls the scale of exploration, and $\zeta_i$ records the visit count at the $i^{th}$ play. *2) Evaluation:* Evaluate the selected leaf $\boldsymbol{z}_\tau$ with the reward model $p_r(r_\tau | \boldsymbol{z}_\tau, g_t)$. *3) Expansion:* Expand the leaf node by adding child nodes. *4) Back Up:* Update the action-values: $Q_i = (Q_{i-1}\zeta_{i-1} + r_\tau)/(\zeta_{i-1} + 1)$ and increment the visit count: $\zeta_i = \zeta_{i-1} + 1$ on all the traversed edges.

**Dyna-Q + MCTS** initializes Q values in MCTS by utilizing the Dyna-Q network. This combination enables MCTS to perform the tree search with more efficiency and efficacy (Silver et al., 2016).

## 3 OBJECT-ORIENTED DYNAMICS MODELS

We introduce the Object-Oriented Text Dynamics (OOTD) models, including 1) a transition model that directly predicts the belief states of objects and 2) a reward model that maps sampled object states to rewards. Based on these models, the planner determines an action purely based on predicted beliefs and calibrates its belief with the feedback (observations and rewards) from the environment. We introduce the object-supervised and self-supervised objectives to train these models.

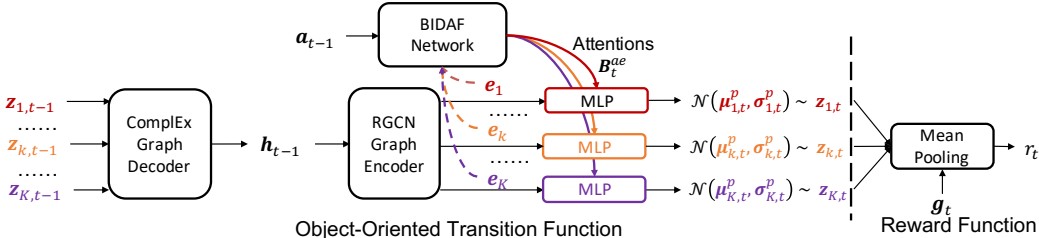

Figure 2: *Left*: The Object-Oriented Transition Model. *Right:* The Reward Model.

### 3.1 OBJECT ORIENTED TRANSITION MODEL

Figure 2 illustrates our transition model, which has three major components *(a)* a graph decoder, *(b)* a graph encoder, and *(c)* independent transition layers.

*(a)* **Graph Decoder.** Given the states of $K$ objects $\boldsymbol{z}_{t-1} = [z_{1,t-1}, ..., z_{K,t-1}]$, we implement a graph decoder with the ComplEx scoring function (Trouillon et al., 2016) that maps the object states to a memory graph $h_{t-1} \in [0, 1]^{C \times K \times K}$. The scoring function for an entry $(c, i, j)$ (in the adjacency matrix of $h_{t-1}$) is implemented by $Re(\langle \omega_c, z_{i,t-1}, z_{j,t-1} \rangle)$ where $\omega_c \in \mathbb{C}^E$ is a complex vector for

the relation $c$ and $Re$ denotes the real part of the decomposition in complex space. The function can model a large variety of binary relations, including both symmetric and antisymmetric relations. Based on this scoring function, predicting $h_{t-1}$ is equivalent to estimating the probability of having a relation $c$ between object $i$ and $j$, which can be efficiently approximated with a low-ranked ($E \ll K$) matrix decomposition: $h_t = [Re(\boldsymbol{Z}\boldsymbol{W}_1\boldsymbol{Z}^T), ..., Re(\boldsymbol{Z}\boldsymbol{W}_C\boldsymbol{Z}^T)]$ where $\boldsymbol{Z} \in \mathbb{R}^{K \times E}$ is the matrix of object states and $\boldsymbol{W}_c \in \mathbb{C}^{E \times E}$ is a complex matrix. In practice, this decomposition enables to map efficiently from $\boldsymbol{z}_{t-1}$ to $h_{t-1}$, therefore scaling our method to environments with numerous objects.

*(b)* **Graph Encoder.** We encode the memory graph $h_{t-1}$ into node representations $\boldsymbol{e}_{t-1} = [e_{1,t-1}, ..., e_{K,t-1}]$ with a Relational Graph Convolution Network (R-GCN) (Schlichtkrull et al., 2018). Nodes in $h_t$ correspond to candidate objects and RGCN performs message passing between nodes, so $e_{k,t-1}$ captures the semantic information of the $k^{th}$ object and its relations to other nodes (objects). Compared to $\boldsymbol{z}_{t-1}$, $\boldsymbol{e}_{t-1}$ denotes the object embeddings after taking into account relations to other objects in the graph. The graph can be thought as a succinct latent representation that ignores irrelevant object information in text observations when predicting the next state $\boldsymbol{z}_t$.

*(c)* **Independent Transition Layers.** TBG environments typically contain a large population of objects, and each action can only affect a few underlying objects. In order to improve the robustness of the dynamics, we utilize 1) a Bi-Directional Attention Flow (BIDAF) network (Seo et al., 2017) to identify the affected objects and 2) a group of transition layers to predict the belief state of objects by following the Independent Causal Mechanism (ICM) framework (Pearl, 2009).

Given $\boldsymbol{e}_{t-1}$ and an action $\boldsymbol{a}_{t-1} = [a_{1,t-1}, ..., a_{J,t-1}]$ (a sentence of $J$ words), the BIDAF network learns an attention matrix $\boldsymbol{B}^{ae} \in [0, 1]^{K \times J}$, where $b_{k,j}$ measures the similarity between the representation of the $k$-th object and the embedding of the $j$-th word. To learn an *action-aware representation of object*, we compute 1) *action-to-object attentions* $\boldsymbol{b}^e$ that indicate which object representations have the closest similarity to one of the action words and are hence critical for modelling the impact of the action words; 2) *object-to-action attentions* $\boldsymbol{b}_k^a$ that indicate which action words are most relevant to the $k^{th}$ object representation. The action-aware object representation concatenates the attended action vector $\boldsymbol{\nu}_{k,t-1}^a$, the attended object representation $\boldsymbol{\nu}_{t-1}^e$ and the initial object representation $\boldsymbol{e}_{k,t-1}$:

$$\boldsymbol{\nu}_{k,t-1}^a = \sum_j b_{k,j}^a \psi^a(a_{j,t-1}) \quad \text{where} \quad \boldsymbol{b}_k^a = softmax(\boldsymbol{B}_{k,:}^{ae}) \in [0, 1]^J,$$

$$\boldsymbol{\nu}_{t-1}^e = \sum_k b_k^e \boldsymbol{e}_{k,t-1} \quad \text{where} \quad \boldsymbol{b}^e = softmax(\max_{col}(\boldsymbol{B}^{ae})) \in [0, 1]^K,$$

$$p_{\mathcal{T}}^k(z_{k,t}|\boldsymbol{a}_{t-1}, \boldsymbol{z}_{t-1}) = \mathcal{N}(\boldsymbol{\mu}_{k,t}, \boldsymbol{\sigma}_{k,t}) \quad \text{where} \quad [\boldsymbol{\mu}_{k,t}, \boldsymbol{\sigma}_{k,t}] = \psi_k^p([\boldsymbol{\nu}_{k,t-1}^a, \boldsymbol{\nu}_{t-1}^e, \boldsymbol{e}_{k,t-1}]) \quad (3)$$

Here $\psi^a$ is a transformer-based text embedding function (Vaswani et al., 2017) for the input action, and $\psi_k^p$ is implemented by a MLP. The transition layers $\psi_1^p, ..., \psi_K^p$ are parameterized independently.

## 3.2    OBJECT ORIENTED REWARD MODEL

We learn a reward model $p_r(r_t|\boldsymbol{z_t}, g_t)$ based on the predicted belief states. This reward model is important for learning goal-conditioned policies that can utilize the knowledge learned from a task to solve similar tasks, which accelerates learning and facilitates knowledge generalization.

To implement the reward model, we build a goal extractor (Appendix A.3) that extracts goals $g_t$ from object states $\boldsymbol{z}_t = [z_{1,t}, ..., z_{K,t}]$. The goal is described by a sequence of words and symbols that follow the triplet format, e.g., "[potato, -, need, -, fry]" where "-" serves as separator between object and relation mentions. Although $\boldsymbol{z}_t$ implicitly embeds $g_t$ with latent values, we prefer to represent $g_t$ explicitly with a sentence for the ease of model comprehension. Given the goal $g_t$ and object states $\boldsymbol{z}_t$, the reward model $p_r$ is defined by

$$p_r(r_t|\boldsymbol{z}_t, g_t) = \psi^r[\boldsymbol{z}_t^{pool}, \psi^g(g_t)] \text{ where } \boldsymbol{z}_t^{pool} = mean\_pooling[z_{1,t}, ..., z_{K,t}], \quad (4)$$

$\psi^g$ is a transformer-based text embedding function for goals, and $\psi^r$ is implemented by a MLP.

## 3.3    TRAINING OBJECTIVE FOR DYNAMICS MODELS

In this section, we introduce the objectives for learning the transition model and the reward model under the object-supervised setting and the self-supervised setting.

**Object-Supervised Setting.** We learn the dynamics model under the supervision of memory graphs $h_t$ that record rich information about the observed objects and their relations from the beginning of a

game (Section 2.3). The Object-Supervised Evidence Lower Bound (OS-ELBo) objective is:

$$\sum_{t=1}^{T} \left\{ \mathbb{E}_{q_{\mathcal{E}}} \left[ \log p_{\Omega}(\hat{h}_t | \boldsymbol{z}_t) \right] - \sum_{k=1}^{K} D_{kl} \left[ q_{\mathcal{E}}^{k}(z_{k,t}|o_t, a_{t-1}, \boldsymbol{z}_{t-1}) \| p_{\mathcal{T}}^{k}(z_{k,t}|a_{t-1}, \boldsymbol{z}_{t-1}) \right] \right\} \quad (5)$$

where $p_{\Omega}$ is the graph decoder, $p_{\mathcal{T}}$ is the transition model and $q_{\mathcal{E}}^{k}$ is the observation encoder. We now provide more details about each component and how to obtain $h_t$. $p_{\Omega}$ is approximated by a re-sampling technique described in Appendix A.2. $p_{\mathcal{T}}$ is a Gaussian distribution as described in Equation 3. $q_{\mathcal{E}}^{k}$ shares a similar structure to $p_{\mathcal{T}}$ (see Appendix A.5 for more details).

The memory graph $h_t$ is not a direct output from the RL environment. To perform object-supervised training, we extract object information from $o_t$ and predict $\hat{h}_t$ with a deterministic object extractor $f_e(o_t, a_{t-1}, h_{t-1})$. We learn $f_e$ during a *pre-training* stage by utilizing a FTWP dataset. The dataset records the trajectories $[o_1, a_1, \ldots, o_t, a_t]$ and corresponding $[h_1, \ldots, h_t]$ by following walkthroughs in FTWP games. In practice, we can build such a dataset by extracting triplets ($\phi$) from the text observations ($o_1, \ldots, o_t$) with an Open Information Extractor (OIE) (Angeli et al., 2015) and map the captured triplets into a knowledge graph by following (Ammanabrolu & Hausknecht, 2020). After training $f_e$, we initialize $h_0 = \{0\}^{C \times K \times K}$ and update by $\hat{h}_t = f_e(o_t, a_{t-1}, \hat{h}_{t-1})$ during *training*. The detailed introduction of $f_e$ and the FTWP dataset can be found in Appendix A.4 and A.6.

To train the reward model, we sample object states $\boldsymbol{z}_t = [z_{1,t}, \ldots, z_{K,t}]$ from their belief states, extract goal $g_t$, and minimize a smooth L1 loss (Girshick, 2015):

$$\begin{cases} \frac{1}{2\beta} [p_r(r_t | \boldsymbol{z}_t, g_t) - r_t]^2, & \text{if } |p_r(r_t | \boldsymbol{z}_t, g_t) - r_t| < \beta, \\ |p_r(r_t | \boldsymbol{z}_t, g_t) - r_t| - \frac{1}{2}\beta, & \text{otherwise.} \end{cases} \quad (6)$$

**Self-Supervised Setting.** In order to generalize our dynamics model to common RL environments where the predicted or ground truth memory graphs are unavailable, we develop a Self-Supervised Evidence Lower Bound (SS-ELBo) objective that directly learns from rewards and observations:

$$\sum_{t=1}^{T} \left\{ \mathbb{E}_{q_{\mathcal{E}}} \left[ \log p_o(o_t | \boldsymbol{z}_t) + \log p_r(r_t | \boldsymbol{z}_t, g_t) \right] - \sum_{k=1}^{K} D_{kl} \left[ q_{\mathcal{E}}(z_{k,t}|o_t, a_{t-1}, \boldsymbol{z}_{t-1}) \| p_{\mathcal{T}}(z_{k,t}|a_{t-1}, \boldsymbol{z}_{t-1}) \right] \right\}$$
$$(7)$$

where $p_{\mathcal{T}}$, $p_r$ and $q_{\mathcal{E}}$ are the aforementioned transition model, reward model and observation encoder. The observation decoder $p_o$ is implemented by a Sequence-to-Sequence (Seq2Seq) model (Sutskever et al., 2014), which is trained by a teacher-forcing technique with text observations $\boldsymbol{o}_t$. This SS-EBLo objective enables training the reward and transition model together by maximizing one objective, which improves the training efficiency. Appendix C.1 shows an example of a predicted graph.

## 4 EXPERIMENTS

**Environments.** We divide the games in the Text-World benchmark (Côté et al., 2018) into five subsets according to their difficulty levels. Each subset contains 100 training, 20 validation, and 20 testing games. These subsets are mutually exclusive. For the easier cooking games, the recipe requires a single ingredient and the world is limited to a single location, whereas harder games require an agent to navigate a map of multiple locations to collect and appropriately process up to three ingredients. Table 4 summarizes the game statistics.

**Running Settings.** To efficiently collect samples from the games *in the training dataset*, we *train* our OOTD model by implementing the Dyna-Q algorithm (Algorithm 1 in Appendix) due to the fact that 1) Q-network supports batch input, which enables solving and gathering data from multiple games in parallel, and 2) the Q-based algorithms provide a fair comparison of sample efficiency with other model-free baselines. After training, we fix the parameters in dynamics models and predict belief states and rewards to solve the games in the *test dataset*. During *testing*, we study the options of applying different planning algorithms including Dyna-Q, MCTS, and their combination (Dyna-Q+MCTS) that initializes MCTS with values from Q networks (Algorithm 2 in appendix). The model hyper-parameters are tuned with the games in the *validation* set. (Appendix A.1).

**Baselines.** we compare the state-of-the-art models for solving TBGs. The baselines include Deep Q-Network (**DQN**) (Narasimhan et al., 2015) and Deep Recurrent Q-Network (**DRQN**) (Yuan et al., 2018). DQN selects the action based on the current observation, whereas DRQN models the history of action and observations with an RNN. For fair comparisons, we replace their LSTM-based text

encoders with transformers. We compare an extension of DRQN by using an episodic counting bonus to encourage exploration, which is denoted as (**DRQN+**) (Yuan et al., 2018). Our next baseline is **KG-A2C** (Ammanabrolu & Hausknecht, 2020), which builds a graph-based state space and applies the Actor-Critic algorithm for learning the policy. We replace their action generator with an action selector for comparison. The last baseline is GATA (Adhikari et al., 2020) that learns a graph-structured representation to model the environment dynamics and select actions based on this representation. We experiment with several options of GATA, including: 1) **GATA-GTP** that pre-trains the discrete graph updater with ground-truth graphs from the FTWP dataset (Appendix A.6). Note that we apply the same pre-training dataset for our OS-ELBo objective. 2) **GATA-OG** that learns a continuous graph updater by reconstructing the text observation. 3) **GATA-COC** that trains the graph updater with a contrastive observation classification objective. Both GATA-OG and GATA-COC follow the self-supervised setting, which offers fair comparison to our SS-ELBo objective.

Table 2: Normalized testing scores and averaged improvement ($\uparrow$) over DQN in six difficulty levels (0 to 5). For each method, we implement three independent runs (check random seeds in Appendix A.1). We select the top-performing agents in *validation* games and report their *testing* scores.

| Type | Model | 0 | 1 | 2 | 3 | 4 | 5 | $\uparrow$ |
|---|---|---|---|---|---|---|---|---|
| Model-Free Algorithm | DQN | 90.0 | 62.5 | 32.0 | 38.3 | 17.7 | 34.6 | 0 |
| | DRQN | 95.0 | 58.8 | 31.0 | 36.7 | 21.4 | 27.4 | -0.8 |
| | DRQN+ | 95.0 | 58.8 | 33.0 | 33.3 | 19.5 | 30.6 | -0.8 |
| | KG-A2C | 96.7 | 55.5 | 31.0 | 54.3 | 26.8 | 30.1 | +3.2 |
| | GATA-GTP | 95.0 | 62.5 | 32.0 | 51.7 | 21.8 | 23.5 | +1.9 |
| | GATA-OG | 100 | 66.2 | 36.0 | 58.3 | 14.1 | 45.0 | +7.4 |
| | GATA-COC | 96.7 | 62.5 | 33.0 | 46.7 | 25.9 | 33.4 | +3.9 |
| Model-Based Planning | OOTD learned by the Object-Supervised (OS) ELBo Objective | | | | | | | |
| | OS-Dyna-Q | 100 | 62.5 | 42.0 | 58.3 | 21.8 | 48.2 | +9.6 |
| | OS-MCTS | 95.0 | 77.5 | 56.0 | 63.3 | 24.9 | 42.9 | +14.1 |
| | OS-Dyna-Q + MCTS | 95.0 | 78.8 | **57.0** | 71.7 | 27.7 | 38.1 | +15.5 |
| | OOTD learned by the Self-Supervised (SS) ELBo Objective | | | | | | | |
| | SS-Dyna-Q | 100 | 62.5 | 48.0 | 53.3 | 30.5 | 47.0 | +11.0 |
| | SS-MCTS | 100 | 70.0 | 51.0 | 70.0 | 27.3 | 54.4 | +16.3 |
| | SS-Dyna-Q + MCTS | 100 | **81.3** | 56.9 | **75.0** | **31.4** | **58.4** | **+21.3** |

## 4.1 CONTROLLING PERFORMANCE IN TBG ENVIRONMENTS

Table 2 shows the agent's controlling performance at different difficulty levels. Equipped with the OOTD model, our agent implements model-based planning and achieves leading performance over the agents based on model-free algorithms. We find the SS-based OOTD models outperform the OS-based models by scoring more rewards at 4 (out of 6) difficulty levels. Although the object signals facilitate a better prediction of dynamics, this advantage can not be directly transferred to controlling performance. This is commonly known as the *objective-mismatch* problem in model-based RL (Lambert et al., 2020). On the other hand, SS-based models are directly optimized toward generating a better prediction of rewards, which are important signals for controlling. The reward-irrelevant information is filtered to derive a more compressed representation for better generalization ability. Another important finding is that our OOTD-based agents perform better than other graph-based agents (e.g., GATA-GTP and KG-A2C). This is because OOTD is doing model-based RL by learning the dynamics at the object level while other graph-based techniques are model free since they learn to extract object representations, but not to predict their dynamics.

**Sample Efficiency.** We empirically demonstrate the sample efficiency of our model-based agents by comparing their training performance with other model-free agents. This experiment studies three variations of proposed methods, including 1) OS-Dyna-Q and 2) SS-Dyna-Q that implement the Q-Dyna algorithm and train the OOTD models with the OS-ELBo objective and the SS-ELBo objective respectively. 3) Object-Oriented (OO)-DQN that trains the OOTD model to predict object states, based on which we compute action values. As a model-free method, OO-DQN does *not* expand the replay buffer with the samples generated by dynamics models. Figure 3 shows the training curve for 6 difficulty levels. The learning speed of proposed model-based algorithms (OS-Dyna-Q and SS-Dyna-Q) is generally faster than that of MF algorithms, but we observe an exception at the games of difficulty level 5, where GATA-OG converges faster. We find it generally takes more

samples to learn the interactions among $K$ objects and their relations to action values. It explains why (MF)-Dyna-Q converges slower than others. However, the object-based value function performs better than that based on raw observation in terms of normalized scores.

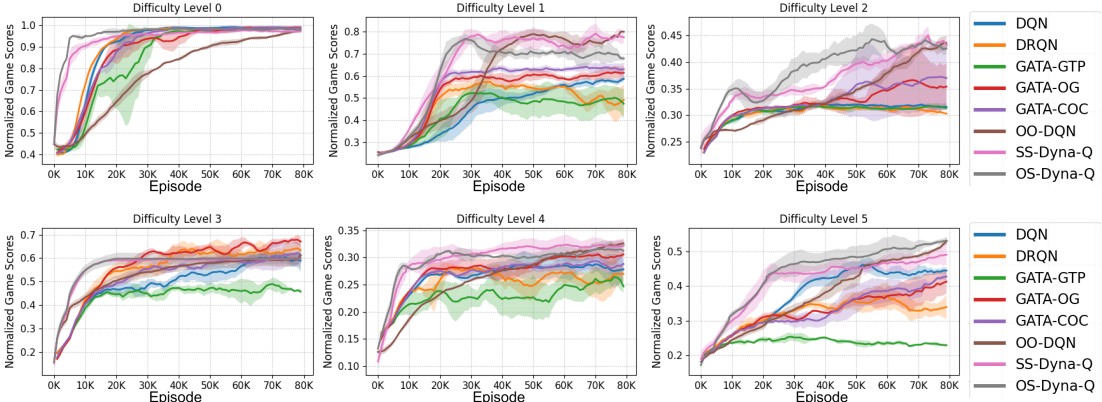

Figure 3: Training Curves: Agents' normalized scores for the games at different difficulty levels. The plot shows $mean \pm std$ normalized scores computed with three independent runs.

## 4.2 ABLATION STUDIES FOR OBJECT-ORIENTED DYNAMICS

To understand the importance of our OOTD design, we empirically study the options of 1) removing the BIDAF attentions by directly applying the action and node embeddings to predict object states (i.e., No-Attentions Dynamics (**NAt-Dyna**)) and 2) removing independent transition layers by building a single layer to predict states for all objects (i.e., Single-Layer Dynamics (**SLa-Dyna**)). We compare these models with our OOTD models trained by the OS-ELBo objective (**OS-OOTD**) and the SS-ELBo objective (**SS-OOTD**). To evaluate how well the modified model $p_{\mathcal{T}'}^k$ captures object dynamics, we sample object states $z_{k,t} \sim p_{\mathcal{T}'}^k(z_{k,t}|a_{t-1}, \boldsymbol{z}_{t-1})$ for a total of $K$ objects, visualize these states with the t-distributed Stochastic Neighbor Embedding (t-SNE) algorithm (Van der Maaten & Hinton, 2008), and quantify the dynamics prediction performance.

**Object States Visualization.** In this experiment, we randomly pick 20 games from the testing games of all difficulty levels. Within each game, $p_{\mathcal{T}'}^k$ computes the states $z_{k,1}, \cdots, z_{k,T}$ with randomly selected actions $a_0, \ldots, a_{T-1}$, and t-SNE embeds $z_{k,t}$ into a vector of 2 dimensions. Figure 4 visualizes the embedded vectors labelled by 10 randomly selected objects. The object states computed by our OOTD model are distinguishable, and thus t-SNE embeds the states for the same object close to each other. This phenomenon is not observable when we apply a single layer for computing object states (i.e., SLa-Dyna). Another important observation is that the OOTD model automatically learns to generate similar states for similar objects. For example, OS-OOTD generates similar states for the object of ingredients (e.g., parsley, apple, carrot, and potato marked by the dark frame). A similar phenomenon, although less obvious, can be observed for SS-OOTD. However, such a shrinkage effect disappears in NAt-Dyna, which explains the importance of bidirectional attentions.

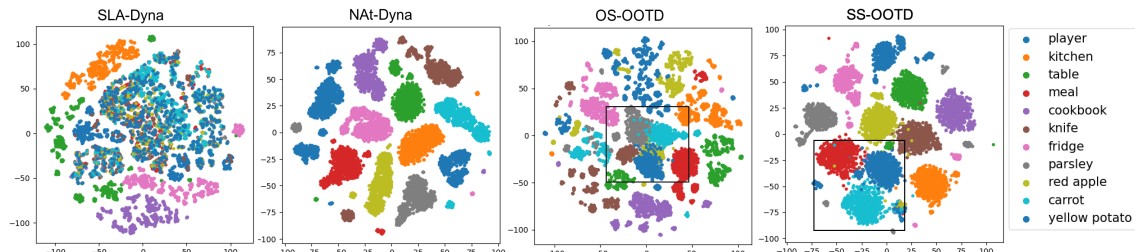

Figure 4: Scatter plots for T-SNE-embedded object states. From left to right, the states are sampled from SLa-Dyna, NAt-Dyna, OS-OOTD and SS-OOTD (from left to right).

**Probing Experiment.** This experiment studies how well our OOTD models can capture useful information from the TBG environments. We map the sampled object states $z_{1,t}, \ldots, z_{K,t}$ to memory

graphs $\hat{h}_t$ and rewards $\hat{r}_t$ with the additional graph decoder $p'_{\mathcal{G}}$ and reward model $p'_r$. The performance of the dynamics models is quantified by whether $\hat{h}_t$ and $\hat{r}_t$ can match the ground-truth $h_t^*$ and $r_t^*$ in the FTWP dataset (Appendix A.6). During sampling, the input actions $a_1, \ldots, a_T$ follow the recorded trajectories in FTWP. The ground-truth rewards in FTWP games is either one or zero, and we perform an entry-level evaluation (e.g., $h_t[i, j, c]$) on the adjacency matrix of a predicted graph. It allows us to evaluate whether the real rewards and triplets are captured with F1 scores. For a fair comparison, the games in the FTWP dataset are divided into training (80%), validation (10%) and testing (10%) sets, based on which we first freeze the parameters in the dynamics models and then update the added $p'_{\mathcal{G}}$ and $p'_r$ with data in training games.

Table 3 shows the testing performance. We study both 1) the graph-*Generation* performance which measures the accuracy of generating all the candidate triplets in a graph and 2) the graph-*Updating* performance which evaluates only the triplets that have been updated at the current step $t$. We add a *Real* and a *Random* baseline that feed the ground-truth graph $h_t^*$ and a randomly generated graph $h_t^{RM}$ into the graph encoder of transition models. They study the probing performance when ground-truth graphs are known and when the posterior collapses for the transition model (so $p_{\mathcal{T}'}^k(\boldsymbol{z}_{k,t}|\boldsymbol{a}_{t-1}, \boldsymbol{z}_{t-1}) = p_{\mathcal{T}'}^k(\boldsymbol{z}_{k,t})$). The results show that OS-OOTD achieves a better graph prediction performance by applying the object-supervised objectives (OS-ELBo), but SS-

Table 3: Graph-generation ($\mathcal{G}$-gen), graph updating ($\mathcal{G}$-upt) and rewards prediction ($r$-pred) performance.

|  | $\mathcal{G}$-gen | $\mathcal{G}$-upt | $r$-pred |
|---|---|---|---|
| Random | 93.5 | 57.4 | 49.5 |
| Real | **99.2** | **95.9** | 94.0 |
| Ablated Dynamics models | | | |
| SLa-Dyna | 84.4 | 75.4 | 85.5 |
| NAt-Dyna | 90.9 | 75.2 | 88.7 |
| OOTD models | | | |
| OS-OOTD | 99.0 | 83.8 | 92.5 |
| SS-OOTD | 77.2 | 65.3 | **94.2** |

OOTD has higher reward prediction accuracy, since SS-ELBo includes an end-to-end training to predict future rewards, which filters the reward-irrelevant object information. Another observation is that removing independent layers and directional attentions significantly degrade the graph prediction performance. It demonstrates that they are important components of our OOTD model.

## 5 RELATED WORK

**DRL Agent for Text-Based Games (TBGs).** TBGs (e.g., Textworld (Côté et al., 2018) and interactive fictional games (Hausknecht et al., 2020)) are interactive simulations with text observations and actions. To solve TBGs, previous DRL agents typically applied advanced neural models for representing the state and action sentences, including recurrent models (Narasimhan et al., 2015; Jain et al., 2020; Zelinka, 2018; Madotto et al., 2020; Adolphs & Hofmann, 2020), Transformers (Adhikari et al., 2020; Ammanabrolu et al., 2020; Urbanek et al., 2019), relevance networks (He et al., 2016), and convolutional neural networks (Yin & May, 2019; Zahavy et al., 2018; Yin & May, 2019). Since states in the TBG environments are partially observable, many previous studies have formulated TBGs as POMDPs and modelled the history of observations with a Deep Recurrent Q-Network (DRQN) (Yuan et al., 2018) or a knowledge graph (Hausknecht et al., 2019; Ammanabrolu & Riedl, 2019; Ammanabrolu & Hausknecht, 2020; Adhikari et al., 2020; Ammanabrolu et al., 2020). Previous works are commonly based on *model-free* Q learning while we explore *model-based* planning.

**Object Oriented Reinforcement Learning.** Diuk et al. (2008) proposed modelling the transition and reward models at the object level. Previous work (Mohan & Laird, 2011; Joshi et al., 2012; da Silva & Costa, 2018; Marom & Rosman, 2018; da Silva et al., 2019) explored training DRL agents with the object-level features. However, object features are not commonly available from the RL environments. Recent works have extracted object representations from states with optical flow (Goel et al., 2018), variational models (Burgess et al., 2019) and structured convolutional layers (Kipf et al., 2020; Finn et al., 2016; Zhu et al., 2018; 2020). Previous works commonly studied *image-based fully observable* environments whereas we explore *text-based partially observable* environments.

## 6 CONCLUSION

We proposed an OOTD model that enables planning in text-based environments. The OOTD model learned a graph representation for capturing object dynamics and predicted the belief of object states with independent transition layers. Empirical results demonstrated that OOTD-based agents outperformed model-free baselines in terms of controlling performance and sample efficiency. We then provided an ablation study to show the importance of OOTD components. A promising future direction will be expanding our OOTD model to parser-based games where the agent must generate the commands character by character without knowing candidate actions.

ACKNOWLEDGMENTS

We acknowledge the funding from the Canada CIFAR AI Chairs program, and the support of the Natural Sciences and Engineering Research Council of Canada (NSERC). Resources used in this work were provided, in part, by the Province of Ontario, the Government of Canada through CIFAR, and companies sponsoring the Vector Institute https://vectorinstitute.ai/partners/.

ETHICS STATEMENT

In this work, we study model-based planning in a simulated (TBG) environment. Although the task itself has limited consequences for society, we take a broader view of our work and discuss its potential influence on future research that might have a considerable impact on our society. We believe developing an autonomous RL agent for TBGs serves as the stepping stone toward real-world applications where the machine cooperates or interacts with humans for some specific tasks. A concrete example of these applications will be the auto-dialogue systems, where the machine must respond to human concerns by conveying accurate information and solving the problem with humans. During this process, failing to communicate clearly, sensibly, or convincingly might also cause harm, for example, without sufficient calibration, the machine could generate improper words or phrases that might potentially cause a bias toward some minorities or underrepresented groups. How to eliminate such bias will be an important direction of future research.

REPRODUCIBILITY STATEMENT

This section introduces how we facilitate the reproducibility from the following perspectives:

**Model Implementation.** Section 4 introduces our approach to dividing the training, validation, and testing datasets as well as the running settings based on the constructed datasets. We report the hyper-parameters for training our OOTD model and the seeds for independent runs in Appendix A.1. The computing resources and running time are discussed in Appendix A.7.

**Code and Dataset.** Please find our implementation on Github[1].

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

## A    MODEL IMPLEMENTATION DETAILS AND DATASETS

### A.1    HYPER-PARAMETERS

We report the key hyper-parameters that have been applied for model implementation. We set the number of candidate objects $K$ to 99 and the number of candidate relations to 10 by following the settings in (Adhikari et al., 2020), which show that the agent can properly solve a game by modelling these objects and relations. The learning rate of policy training and dynamics model training is set to 0.0001. The discount factor $\gamma$ is set to 0.9. The size of the hidden layers and the size of $z$ in our OOTD model are set to 32. The transition layers of different objects are parameterized independently, so the neural network predicts $32 \times 99 = 3,168$ values at each step during dynamics modelling. The sizes of word embedding and node embedding are set to 300 and 100 respectively. These numbers are determined experimentally, which could provide promising performance with a reasonable computational resource. The full list of hyper-parameters is recorded in the $.yaml$ file in our submitted code. In our experiment, the random seeds we select are 123, 321, and 666.

### A.2    RESAMPLING FOR COMPUTING THE GRAPH LIKELIHOOD

The graph likelihood $p_\Omega(h_t|\boldsymbol{z}_t)$ computes the probability of generating $h_t$ with a graph decoder. As it is discussed in Section 3.1, our graph decoder is implemented as a scoring function for entry-level prediction. However, $h_t$ is a graph with significant sparsity, for example, the corresponding adjacency matrix assigns positive labels (i.e., 1) for an average of 97.6 out of all 98,010 entries. As a result, directly optimizing toward $h_t^*$ creates a large negative bias. In other words, the decoder can achieve a satisfying accuracy by labelling all the entries with 0s, which does not capture any object information. To solve this problem, we implement a re-sampling technique that extracts all the positive samples (the entries labelled with one) from $h_t^*$ and sample an equivalent number of negative samples from the rest of the entries. These positive and negative samples are used to estimate the graph likelihood.

### A.3    GOAL EXTRACTOR

At a time step $t$, the goal extractor generates the goal $g_t$ from the object states $\boldsymbol{z}_t$ by optionally conditioning on the previous goal $g_{t-1}$ and observed text $o_t$. The goal can be represented by a sequence of words and symbols that follows the triplet format. To generate this goal, we introduce two kinds of extractors that are built for the OOTD learned by the OS-ELBo and the SS-ELBo objectives.

**Goal Extractor with the OS-ELBo objective.** Under this setting, the predicted memory graph $\hat{h}_t^{OS}$ captures the sparsity of object observations and forms a direct prediction of the ground truth graph $\hat{h}_t^*$ (check Figure 7). We can build a rule-based goal extractor that maps $\boldsymbol{z}_t$ to $\hat{h}_t^{OS}$ and checks whether $h_t^{OS}$ captures certain relations (e.g., 'need') between two objects. If there is such a relation, we directly output the sentence of corresponding triplets by following a format *"[ object 1 $ relation $ object 2 ]"*. Such a simple extractor can generalize well to training and testing games if $\hat{h}_t^{OS}$ manages to capture correct relations.

**Goal Extractor with the SS-ELBo objective.** The memory graph $\hat{h}_t^{OS}$ learned by the SS-ELBo objective captures a latent representation of $\hat{h}_t^*$. This latent representation is not directly interpretable, and thus we build a word-by-word sequence generator whose output is the goal to be predicted ($g_t$) and inputs are $\boldsymbol{z}_t$, $g_{t-1}$ and $o_t$. To train this model, we manually build a goal learning dataset that collects pairs of model input/output from the training games. During testing, we predict the goal after interacting with the environment and update the current goal if a valid goal sentence is generated.

### A.4    OBJECT EXTRACTOR

The object extractor $f_e(o_t, a_{t-1}, h_{t-1})$ predicts the memory graph $h_t$ by conditioning on current observations $o_t$ and action $a_t$. Figure 5 shows the architecture of $f_e$. It shares a similar structure to our OOTD model, but the key differences are 1) $f_e$ is a deterministic model which only maximizes the likelihood of predicting $h_t^*$ during training. 2) $f_e$ captures object information from $o_t$, whereas

$o_t$ is not available for our OOTD model, and it is the motivation why OOTD predicts the belief (distribution) of states while $f_e$ directly generates the deterministic states.

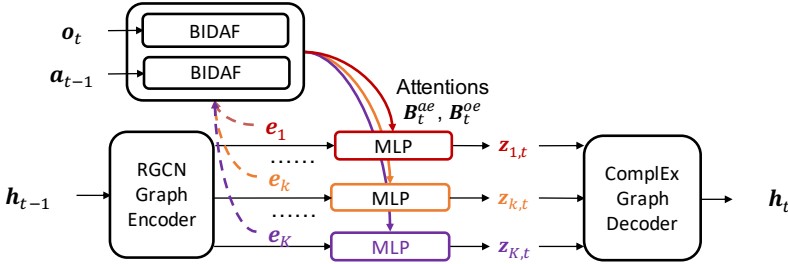

Figure 5: The object extractor.

## A.5 OBSERVATION ENCODER.

The observation encoder $q_{\mathcal{E}}$ is defined by:

$$\boldsymbol{\nu}_{k,t-1}^a = \sum_j b_{k,j}^a \psi^a(a_{j,t-1}) \quad \text{where} \quad \boldsymbol{b}_k^a = softmax(\boldsymbol{B}_{k,:}^{ae}) \in [0,1]^J,$$

$$\boldsymbol{\nu}_{t-1}^e = \sum_k b_k^e e_{k,t-1} \quad \text{where} \quad \boldsymbol{b}^e = softmax(\max_{col}(\boldsymbol{B}^{ae})) \in [0,1]^K,$$

$$\boldsymbol{\nu}_{k,t-1}^o = \sum_j b_{k,j}^o \psi^o(a_{j,t-1}) \quad \text{where} \quad \boldsymbol{b}_k^o = softmax(\boldsymbol{B}_{k,:}^{oe}) \in [0,1]^J$$

$$\boldsymbol{\nu}_{t-1}^{\prime,e} = \sum_k b_k^{oe,e} e_{k,t-1} \quad \text{where} \quad \boldsymbol{b}^{oe,e} = softmax(\max_{col}(\boldsymbol{B}^{oe})) \in [0,1]^K \tag{8}$$

$$q_{\mathcal{E}}^k(z_{k,t}|\boldsymbol{o}_t,\boldsymbol{a}_{t-1},z_{k,t-1}) = \mathcal{N}(\boldsymbol{\mu}_{k,t}^q,\boldsymbol{\sigma}_{k,t}^q) \text{ where } [\boldsymbol{\mu}_{k,t}^q,\boldsymbol{\sigma}_{k,t}^q] = \psi_k^q([\boldsymbol{\nu}_{k,t-1}^o,\boldsymbol{\nu}_{t-1}^{\prime,e},\boldsymbol{\nu}_{k,t-1}^a,\boldsymbol{\nu}_{t-1}^e,e_{k,t-1}])$$

$\psi^o$ is a transformer-based text embedding function (Vaswani et al., 2017) for observations, and $\psi_k^q$ is implemented by a MLP. The transition layers $\psi_1^q, \ldots, \psi_K^q$ are parameterized independently.

Figure 6 shows the structure of an observation encoder $q_{\mathcal{E}}^k$, which shares a similar structure to the transition (OOTD) model (Figure 3), but it includes an additional BIDAF network to learn an attention matrix $\boldsymbol{B}^{oe}$, where $b_{k,i}^{oe}$ measures the similarity between the representation of the $k^{th}$ object and the $i^{th}$ word in the current *observation* $o_t$. It predicts the memory graph $h_t$ from $\boldsymbol{z}_t^q$ with a ComplEx graph decoder (Section 3.1). It predicts the current observation $\hat{o}_t$ and reward $r_t$ with a sequence decoder and the reward function (Section 3.2).

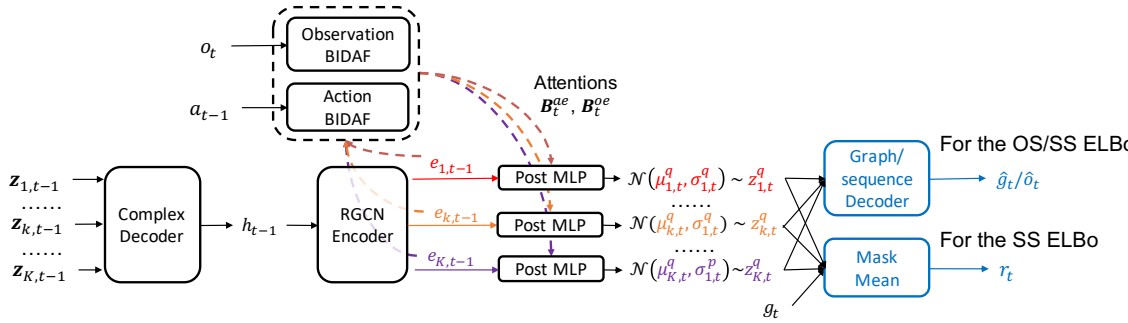

Figure 6: The observation encoder.

## A.6 FTWP DATASET

The FTWP dataset is a public dataset that supports pre-training the dynamics model [2]. Trischler et al. (2019) provided the First TextWorld Problems (FTWP) dataset. The dataset consists of TextWorld

---

[2]The dataset can be downloaded at `https://aka.ms/ftwp/dataset.zip`

games in a cooking theme across a wide range of difficulty levels. For each game, the expert trajectory and the memory graph $h_t$ at each time step are recorded. The original dataset has only 10 games for each difficulty level. This is insufficient for dynamics model pre-training, so Adhikari et al. (2020) further expanded this dataset by adding more games into this dataset by utilizing the logic in the Textworld game engine. The specific generation rules are described in (Adhikari et al., 2020). After expansion, the dataset contains 4440 games for training, 222 games for validation, 514 games for testing. To ensure fairness when using this dataset, we confirm that there is no overlap between the FTWP and the games we used to train and evaluate our planning algorithms.

## A.7 COMPUTATIONAL RESOURCE AND RUNNING TIME

We run the experiment on a cluster operated by the Slurm workload manager. The cluster has multiple kinds of GPUs, including Tesla T4 with 16 GB memory, Tesla P100 with 12 GB memory, and RTX 6000 with 24 GB memory. We used machines with 24 GB of memory for pre-training the object extractor and 64 GB for training the OOTD model. The number of running nodes is 1, and the number of CPUs requested per task is 32. Given the aforementioned resources, pre-training can be completed within 4 days (96 hours), and training the dynamics model takes 6 days (144 hours).

Table 4: Game statistics at different difficulty levels.

| Level | Recipe Size | #Locations | Max Scores | Need Cut | Need Cook | #Action Candidates | #Objects |
|-------|-------------|------------|------------|----------|-----------|--------------------|----------|
| 0 | 1 | 1 | 3 | ✗ | ✗ | 10.5 | 15.4 |
| 1 | 1 | 1 | 4 | ✓ | ✗ | 11.5 | 17.1 |
| 2 | 1 | 1 | 5 | ✓ | ✓ | 11.8 | 17.5 |
| 3 | 1 | 9 | 3 | ✗ | ✗ | 7.2 | 34.1 |
| 4 | 3 | 6 | 11 | ✓ | ✓ | 28.4 | 33.4 |
| 5 | Mixture of Levels[1,2,3,4} | | | | | | |

## B  ALGORITHMS

---
**Algorithm 1:** Dyna-Q Training

---
**Input** : Transition Model: $p_{\mathcal{T}}(z_{t+1}|a_t, z_t)$,
     Object Encoder: $q_{\mathcal{E}}(z_{t+1}|o_{t+1}, a_t, z_t)$,
     Reward Model: $p_r(r_{t+1}|z_{t+1}, a_t, g_t)$,
     Object Extraction function: $f_e(h_t, a_t, o_{t+1})$,
     Graph Decoder: $p_{\Omega}(h_t|z_t)$,
     Training game ids $\mathcal{I}$.
Initialize Replay Buffers $\mathcal{D}$;
**while** *not converged* **do**
  Initialize belief state $z_0$ =random numbers, action $a_0$ ="*restart*", goal $g_0$ ="*<pad>*" and
   memory graph $h_0 = \mathbf{0}^{C \times K \times K}$;
  Randomly pick a game $id \in \mathcal{I}$, set $t = 0$;
  **while** *not end* **do**
    /* Q learning with environment */
    $o_{t+1}, r_{t+1} \leftarrow env(a_t, id)$;
    Extract object information $h_{t+1} = f(a_t, z_t, o_{t+1})$;
    Update object belief $z_{t+1} \sim q_{\mathcal{E}}(z_{t+1}|o_{t+1}, a_t, z_t)$;
    Extract goal: $g_{t+1} = ExtactGoal(z_{t+1}, o_{t+1}, g_t)$;
    Action selection: $\hat{a}_{t+1} = EpsilonGreedy(Q(z_{t+1}, a_{t+1}, g_{t+1}))$;
    Expand the replay buffer: $\mathcal{D} \cup \{z_t, h_t, a_t, g_t, r_{t+1}, z_{t+1}, h_{t+1}, g_{t+1}\}$;
    $t = t + 1$ and $end = CheckEnd(env)$
  **end**
  **for** *update step c=1,...,C* **do**
    /* Model fitting */
    Sample a transition: $\mathcal{T} = \{z_t, h_t, a_t, g_t, r_{t+1}, z_{t+1}, h_{t+1}, g_{t+1}\}$ from the dataset $\mathcal{D}$;
    Update the reward model to minimize the L1Smooth loss (Equation 6) with $\mathcal{T}$;
    Update the transition model to minimize the OS-ELBO (Equation 5) objective or the
     SS-ELBO objective (Equation 7) with $\mathcal{T}$;
    Update the Q function to minimize the TD Loss (Equation 1) with $\mathcal{T}$;
    **for** *explore step b=1,...,B* **do**
      /* Exploration with the transition and reward models */
      Randomly select $a'_{\tau} \in \mathcal{A}$ ;
      $z'_{\tau+1} \sim p_{\mathcal{T}}(z'_{\tau+1}|a'_{\tau}, z_{\tau}), r'_{\tau+1} \sim p_r(r'_{\tau+1}|z'_{\tau+1}, a'_{\tau}, g_{\tau})$;
      Update the transition $\mathcal{T}' = \{z_t, g_t, a'_t, r'_{t+1}, z'_{t+1}, g_{t+1}\}$;
      Update the Q function to minimize the TD Loss (Equation 1) with $\mathcal{T}$;
    **end**
  **end**
**end**

---

---

**Algorithm 2:** MCTS Testing

---

**Input** :Transition Model: $p_{\mathcal{T}}(\boldsymbol{z}_{t+1}|a_t, \boldsymbol{z}_t)$,
   Object Encoder: $q_{\mathcal{E}}(\boldsymbol{z}_{t+1}|o_{t+1}, a_t, \boldsymbol{z}_t)$,
   Reward Model: $p_r(r_{t+1}|\boldsymbol{z}_{t+1}, a_t, g_t)$,
   Action Value function: $Q(\boldsymbol{z}_t, a_t, g_t)$,
   Testing game ids $\mathcal{I}$

Initialize belief state $\boldsymbol{z}_0$ =random numbers, action $a_0$ ="*restart*", and goal $g_0$ ="*<pad>*";
Initialize reward buffer $\mathcal{R}$;
**for** *game id $id \in \mathcal{I}$* **do**
 Initialize root $N_0 = InitTree(Q(\boldsymbol{z}_t, a_t, g_t))$, set $t = 0$ and $R = 0$;
 **while** *not end* **do**
  $o_{t+1}, r_{t+1} \leftarrow env(a_t, id)$ and $R = R + r_{t+1}$;
  Update object belief $\boldsymbol{z}_{t+1} \sim q_{\mathcal{E}}(\boldsymbol{z}_{t+1}|o_{t+1}, a_t, \boldsymbol{z}_t)$;
  Extract goal: $g_{t+1} = ExtactGoal(\boldsymbol{z}_{t+1}, o_{t+1}, g_t)$;
  Select action: $\hat{a}_{t+1} = MCTS\_Simulations(N_t, \boldsymbol{z}_{t+1}, g_{t+1}, p_{\mathcal{T}}(\cdot), p_r(\cdot), Q(\cdot))$;
  Move the search node: $N_{t+1} = Move(N_t, \hat{a}_{t+1})$;
  Expand $t = t + 1$ and $end = CheckEnd(env)$
 **end**
 Buffer total rewards $\mathcal{R} = \mathcal{R} \cup R$
**end**
**output** :Reward buffer $\mathcal{R}$

---

## C COMPLEMENTARY RESULTS

### C.1 GRAPH VISUALIZATION

**Graph Visualization.** Figure 7 illustrates examples of the ground-truth memory graph $h_t^*$ and the graphs learned by the OS-ELBo ($h_t^{OS}$) and SS-ELBo ($h_t^{SS}$) objectives. Since the space of adjacency matrices is large ($K \times K \times C$) while the observed object information is limited, $h_t^*$ can be represent by a sparse matrix. Supervised by predicted graphs $\hat{h}_t$, OS-ELBo captures the sparsity, and $h_t^{OS}$ is a direct approximation to $h_t^*$. However, graph information are unavailable for the SS-ELBo objective, so $h_t^{OS}$ forms a dense latent representation of $h_t^*$. Similar phenomenon can be observed in GATA (Adhikari et al., 2020). We study how well this latent representation and the corresponding object states can represent the object dynamics in Section 4.2.

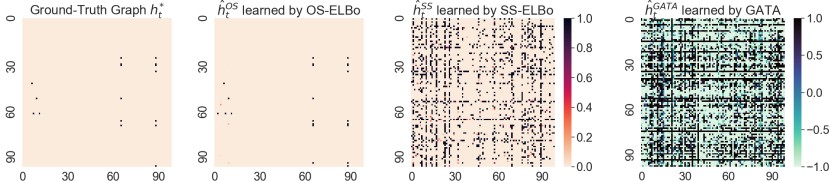

Figure 7: Adjacency tensors corresponding the *"is"* relation in memory graphs $h_t$. From left to right: 1) the ground-truth graph, and the graph learned by 2) the OS-ELBo objective, 3) the SS-ELBo objective (Section 3.3), and 4) GATA (a graph-based dynamics model *baseline*) (Adhikari et al., 2020).

### C.2 PLANNING EFFICIENCY

We compare the efficiency of OOTD-based planning algorithms (including Dyna-Q, MCTS, and Dyna-Q+MCTS) by studying which method can gather more rewards with a limited number of selected actions. Figure 8 shows results. Among the planning algorithms, Dyna-Q+MCTS significantly outperforms others. It is because the Q network provides prior knowledge over the preference of

actions, based on which MCTS implements Monte-Carlo rollouts to generate and evaluate the belief states of objects. Together, they yield a more accurate look ahead of future rewards.

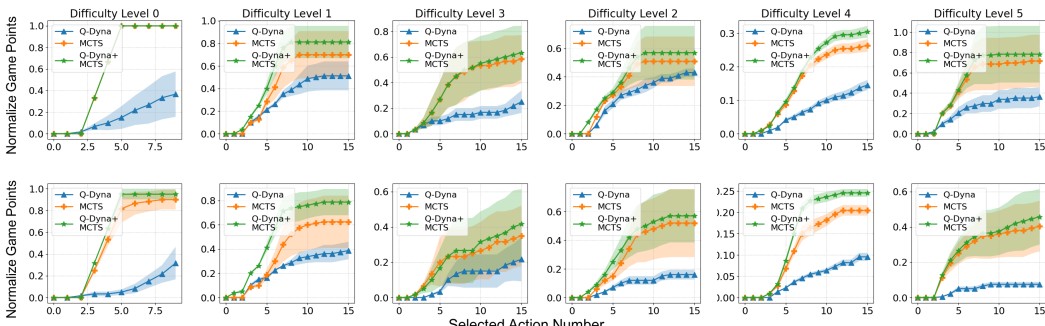

Figure 8: Planning Curve. The agent are based on the dynamics model with the OS-ELBO objective (Upper) and the SS-ELBo objective (Lower). The plot shows $mean \pm std$ normalized scores computed with three independent runs.

