# OpenReview forum: "Learning Object-Oriented Dynamics for Planning from Text"
_ICLR.cc/2022/Conference — ICLR 2022 Poster_

### Official Review · Reviewer_enCT · 2021-11-01

**Correctness:** 4
**Technical Novelty And Significance:** 3
**Empirical Novelty And Significance:** 3
**Recommendation:** 8
**Confidence:** 4

**Main Review:**

Overall I found that the paper is clearly written, with helpful illustrations. My specific questions follow.

**Contributions**

1. I am not sure if "extension of OO-MDP to partial observability" is the contribution of this paper. For example http://h2r1.cs.brown.edu/wp-content/uploads/wandzel19.pdf
2. Abstract: dynamics models for image-based games are primarily on fully observable states -- I'm not sure if this is true. For example https://arxiv.org/pdf/1803.10122.pdf
3. Intro Paragraph 1: generalize to all states and actions -- I don't see how this is exhaustively evaluated in the paper, nor did I see why this is important. Technically, we only need to make predictions about states that we are interested to achieve (and this is also how the authors collected the data).

**Model**

It will be important to showcase or verify that the modeling of belief is important. I think the authors were slightly mixing two things together:
- The partial observability of an environment (e.g., the position of an object is unknown).
- The fact that at each step, the observation-only contains objects with state changes.

As the authors also stated, there is an implicit assumption about "other object states do not change." This is a very strong assumption and can make model learning much easier, as in the STRIPS representation. See also https://en.wikipedia.org/wiki/Frame_problem
Actually, under this assumption, we don't have to treat it as a POMDP (instead, just an MDP). The authors should make the significance of their POMDP formulation more clear.

I would also suggest authors try an ablation study with the belief prediction part removed (i.e., do not use sampling in the transition model).

On the writing of Section 2 and 3, I suggest that authors include ALL important modules in the paper (for example, the goal encoder). Some of the details are just echoing existing work, for example, OO-POMDP, MCTS, and can be potentially shortened. The writing can also be improved by incorporating a concrete running example showing the dynamics and the decision made.

**Summary Of The Paper:**

This paper presents a novel approach for task-based games. The authors used an object-oriented POMDP formulation for the task. The key contribution is to learn the dynamics of the environment with a graph neural network (with either object supervision or self-supervision), and apply an online planning algorithm (e.g., MCTS) to solve the problem. The authors show strong improvements over other approaches in terms of sample efficiency and task scores.

**Summary Of The Review:**

The paper is well-written and the results are strong. I have a few comments on the formulation and presentation of the paper. I also suggested an ablation study.

---

> ### Author Response · Authors · 2021-11-14
> **Response to Reviewer enCT**
>
> Dear Reviewer, we appreciate your constructive comments and provide our response to your concerns as follows.
>
> 1. *"I am not sure if "extension of OO-MDP to partial observability" is the contribution of this paper."*
> - ***Response:*** Our model design is from different that of their model. Unlike [wandzel19] that represents object states with attributes and symbolic features, we develop a latent object representation that allows our OOPOMDP generalize to the complex environment based on high-dimensional features (e.g., images and text). We have clarified it in the revised version.
>
> 2. *"Abstract: dynamics models for image-based games are primarily on fully observable states. -- I'm not sure if this is true."*
> - ***Response:*** We will change "Existing dynamics models commonly" to "Dynamics models mostly" to make this statement more precise and consistent with our introduction.
>
> 3. *"Intro Paragraph 1: generalize to all states and actions -- I don't see how this is exhaustively evaluated in the paper, nor did I see why this is important. Technically, we only need to make predictions about states that we are interested to achieve."*
> - ***Response:*** Yes, we did not mean to generalize to all states and actions, but simply to generalize at test time.  We rephrased the sentence accordingly.
>
> 4. *"As the authors also stated, there is an implicit assumption about "other object states do not change."
> Actually, under this assumption, we don't have to treat it as a POMDP (instead, just an MDP). The authors should make the significance of their POMDP formulation more clear."*
> - ***Response:*** We assume "an input action influences the states of a limited number of objects", but we do not know which objects have been influenced. The influenced objects might not be included in the current observation, and we need to locate them with history data.  For example, the agent was in the "bedroom", and it performs an action "go east" to enter the "kitchen". The current observation will only describe the kitchen. However, the state of "bedroom" will be influenced since player is no longer in the "bedroom". The agent must infer this relation by remembering "player was in bedroom", so we need to use a POMDP in this case.
>
> 5. *"I would also suggest authors try an ablation study with the belief prediction part removed (i.e., do not use sampling in the transition model)."*
> - ***Response:*** We compare to GATA, which is a baseline that infers the current belief, but does not predict future beliefs and therefore does not do any planning.

---

### Official Review · Reviewer_cFdN · 2021-11-02

**Correctness:** 4
**Technical Novelty And Significance:** 3
**Empirical Novelty And Significance:** 3
**Recommendation:** 8
**Confidence:** 3

**Main Review:**

# strengths and weaknesses
- strengths
  - Tackles a difficult problem
  - Presents a useful extension to OO POMDPs which can be used to model many text-based games
  - Presents good evidence that model-based training using their learned dynamics model is an improvement over the baseline (model-free learning)
  - Achievements in the domain of text-based games are important since they could lead to better conversational agents, which may be deployed in complex settings that require tracking conversation history, e.g., in IT, health
- weaknesses
  - Some technical details could use more explanation, which would improve reproducability and increase my confidence in the technical soundness of the work (see comments)
  - Although there are comparisons to model-free methods, it seems that a comparison with a model-based method is missing (table 2)? Could the authors comment on how they expect their OOTD approach to compare to, e.g., a simple RNN that takes text observations as input and outputs a list of object states. Are there no model-based baselines, since this work is the first in the text-based games domain?

# Questions for clarification
- sec 2.2: Does $S$ contain all the information of $Z$? Is there a one-to-one correspondence between $S$ and $Z$? Is $Z$ completely recoverable from $S$?
- sec 2.2: What exactly is contained in the object level states $Z$? You extend Diuk et al 2019, which defines a set of object classes and attributes for each class. Are classes and attributes represented explictly anywhere in your framework? Or are the object states $Z$ hidden state vectors that are not constrained to take any particular form?
- sec 2.4: In equation 2, why isnt $\zeta$ indexed by the goal $g$? Shouldn't the tree-search nodes record the current goal?
- sec 3.1 (a): I think $Re(\omega_c, z_{i, t-1}, z_{j, t-1})$ needs something in between the parentheses. Perhaps something like $Re(f_{score}(\omega_c, z_{i,t-1}, z_{j,t-1}))$? Otherwise, do you mean that $Re()$ is being applied to the concatenation of three vectors?
- sec 3.1 (c): How big is the space of actions? It sounds like each action is identified by a (possibly lengthy) sentence. Did you have any trouble with exploration?
- sec 3.3 and A.3: What happens if the goal extractor finds multiple plausible goals in the memory graph? What if nothing valid is found?
- sec 4 table 2: why aren't the GATA baselines in the "model based planning category"? Don't they also model system dynamics?
- sec 3.3 equations 5 and 7: Can you explain how you arrive at your objective? I'm familiar with one way that evidence lower bound can appear: $\log P(X) - D_{KL}(Q || P)$. Can you give some insight into why the first term in equations 5 and 7 involves an expectation?
- sec 3.3 equations 5 and 7: Similarly, in the second term, can you explain the ordering of arguments in $D_{KL}(q_{\epsilon} || p_{\mathcal{T}})$ ? I would expect that since, $p_{\mathcal{T}}$ is the function being learned, it should be the first argument in the KL divergence, as is typically the case when using an ELBo objective.

# Typos
- pg 2. typo? "to enable stochasticity in the dynamics" --> "to enable stochacisity in the training"
- pg 6. typo: "is implement by" --> "is implemented by"
- pg. 7: "compassion" --> "comparison"
- pg. 7 bottom: sentence beginning with "3)" seems incomplete
- pg. 8: "object closed to each other" --> "close"
- pg. 8: figure 4 -- last two subfigures have same label

**Summary Of The Paper:**

This paper tackles the difficult problem of learning to play text-based games. This domain is particularly challenging, since the text observations received at each time-step are variable length, and will only provide a partial description of the world, from which the full state must be reconstructed. Additionally, the player must track the history of past states in order to make the correct decision.

This paper introduces a new framework for describing the time-evolution of a game: Object-Oriented Partially Observable Markov Decision Processes (OO POMDP). This work's main contribution is a learned Object-Oriented Text Dynamics (OOTD) model that learns the transition and reward function of a OO POMDP game.

Briefly, the presented method for predicting the transition function is: given a representation $z_t$ of the objects and $a_t$ of the action, create a graph, where the nodes are objects and the edges denote relationships. Perform message passing, and obtain new node representations $e_t$.

Then, (1) given $e_t$, predict a new representation $v_{t}$ using dual stream attention with the action $a_t$. Then, (2) use $v_t$ to predict the updated state $z_{t+1}$ of each object, using an independent set of parameters per object.

The dynamics model is trained using a Evidence lower bound objective (ELBo) in two varieties: one that relies on a deterministic extraction of the graph and one that learns directly from rewards and observations.

To show its effectiveness, the learned dynamics model is used to model-based training (Dyna-Q, MCTS) is used to learn a planner. The learned planner is shown to out-perform the baselines (DQN, DRQN, GATA), both on the test cases, and in terms of sample efficiency. Ablations of (1) and (2) find qualitative advantages of the proposed method, since the learned object representation are both more separable and better semantically clustered. Additionally, the ablations are shown to reduce the accuracy of the graphs and state recreated from the object representations.

**Summary Of The Review:**

Overall, I thought this paper was well written and presents a significant contribution in a novel area: modeling dynamics for text-based games. And the proposed extension to OO POMDPs will be generally useful for anyone working on text-based games. The given ablation studies and comparison to model-free baselines provides believable evidence for the effectiveness of their model-based learning approach. And while this work focused mainly on games, it's clear that their results could have useful applications in the field of conversational agents, which is growing in importance. I had some questions about the technical details of their method, especially the formulation of their objective. But these could probably be well answered in follow up discussion or in an appendix.

---

> ### Author Response · Authors · 2021-11-14
> **Response to Reviewer cFdN -- Part1**
>
> Dear Reviewer, we appreciate your constructive comments and provide our response to your concerns as follows.
>
> 1. *"Although there are comparisons to model-free methods, it seems that a comparison with a model-based method is missing (table 2)? ...Are there no model-based baselines, since this work is the first in the text-based games domain?"*
> - ***Response:***  To the best of our knowledge, it is the first work that implements model-based RL in text-based games, and this is one of our contributions. We compared the performance of different planners (e.g., Dyna-Q and MCTS) based on OOTD, and we believe they can be each other's baseline.
>
> 2. *"sec 2.2: Does S contain all the information of Z? Is there a one-to-one correspondence between S and Z ? Is Z completely recoverable from S ?"*
> - ***Response:***  $s_t$ and $z_t$ are not the same.  $s_t$ contains all the information of $z_t$, but not the other way around.  You can think of $s_t$ as an embedding of all past actions and observations that capture all the details of the text in the game.  For instance, $s_t$ could be the embedding of the text produced by an auto-encoder that ensures no loss of information.  In contrast, $z_t$ contains only object level information.  In our model, $z_t$ is a set of embeddings (one per object) that does not contain irrelevant background information.  For example, $s_t$ would contain all the information of "You see a gleam in the corner where you can see a bbq." whereas $z_t$ would only contain information about the location of the bbq.
>
> 3. *"sec 2.2: What exactly is contained in the object level states ? You extend Diuk et al 2019, which defines a set of object classes and attributes for each class. Are classes and attributes represented explicitly anywhere in your framework? Or are the object states Z hidden state vectors that are not constrained to take any particular form?"*
> - ***Response:*** Unlike (Diuk et al 2019) that assigns symbolic features to an object, we learn a latent representation of an object. The dimension of a latent vector is 32. At each step, we model 99 objects, so the total number of latent features is 99*32=3168 at each time step (see Appendix A.1).
>
> 4. *"sec 2.4: In equation 2, why isn't $\zeta$ indexed by the goal ? Shouldn't the tree-search nodes record the current goal?"*
> - ***Response:*** The visit count should be labelled by a goal. we have added it.
>
> 5. *"sec 3.1 (a): I think $Re(\dots)$ needs something in between the parentheses. Otherwise, do you mean that $Re(\dots)$ is being applied to the concatenation of three vectors?"*
> - ***Response:*** You are right, it should be $Re(<\omega_{c},z_{i,t-1},z_{j,t-1}>)=Re(\sum_{e=1}^{E}\omega_{e,c},z_{e,i,t-1},z_{e,j,t-1})$.  We have changed Equation 2 accordingly in the paper.
>
> 6. *"sec 3.1 (c): How big is the space of actions? It sounds like each action is identified by a (possibly lengthy) sentence. Did you have any trouble with exploration?"*
> - ***Response:*** In this work, we study ***choice-based games***, where the candidate commands (or actions) $A_t$ are available and the planner determines the action at $a\in A_t$ to be performed (we mentioned it in Section 2.2 and 2.4). The average number of actions at each time step for each level of the game is reported in Table 4.  For future work, we will study parser-based games where the agent must generate the commands character by character without knowing candidate actions (see Section 6).
>
> 7. *"sec 3.3 and A.3: What happens if the goal extractor finds multiple plausible goals in the memory graph? What if nothing valid is found?"*
> - ***Response:*** For the first question, we will plan for each goal. It commonly happens in the game of difficulty level 4 where we need to gather multiple ingredients (where gathering each ingredient is a separate goal). If the goal extractor does not return any goal, then the agent abandons since it has nothing to plan for and the game ends.
>
> 8. *"sec 4 table 2: why aren't the GATA baselines in the "model based planning category"? Don't they also model system dynamics?"*
> - ***Response:*** GATA is essentially an auto-encoder that does not model any dynamics. The target is learning a representation for states. Nothing is predicted forward. They implement classic Q-learning based on the learned states.
>
> 9. *"sec 3.3 equations 5 and 7: Can you explain how you arrive at your objective? ... Can you give some insight into why the first term in equations 5 and 7 involves an expectation?"*
> - ***Response:*** (5) and (7) are ELBOs for conditional priors and posteriors. The first terms of (5) and (7) are implemented by likelihood loss, which requires the expectation.

---

> > ### Author Response · Authors · 2021-11-14
> > **Response to Reviewer cFdN -- Part2**
> >
> > 10. *"sec 3.3 equations 5 and 7: Similarly, in the second term, can you explain the ordering of arguments? I would expect that since, $p_{\mathcal{T}}$ is the function being learned, it should be the first argument in the KL divergence, as is typically the case when using an ELBo objective."*
> > - ***Response:*** We keep this order since the transition model $p_{\mathcal{T}}$ is the conditional prior (without knowing $o_{t}$) while the observation encoder $q_{\mathcal{E}}$ is the conditional approximate posterior (after observing $o_{t}$). Both $p_{\mathcal{T}}$ and $q_{\mathcal{E}}$ are to be learned. We believe the order of prior and posterior should be correct in the KLD term.
> >
> > 11. *"Typos"*
> > - ***Response:*** We have fixed them.

---

> > > ### Comment · Reviewer_cFdN · 2021-11-29
> > > **acknowledgement of author response**
> > >
> > > # acknowledgement of author response
> > > I thank the authors for their clear and thorough responses. The responses have slightly increased my confidence in the technical soundness of the work and its reproducibility, and my score remains the same:  accept.
> > >
> > > I agree with reviewer W88S that more explanation for $q_{\epsilon}$ is needed. The updates to section 3.3 clarify where the memory graph $h_t$ comes from, but I still believe that an explanation of the loss function is needed. Important improvements that would be helpful include:
> > > - Clarifying what the output of training is: Is it $q_\epsilon$ or $q_{\mathcal{T}}$? Both of them give a posterior distribution over states $z$ and it seems to me that $q_{\mathcal{T}}$ is redundant, since at evaluation time, we will always want to condition on the observation $o$, and thus will want to use $q_\epsilon$ for model updates.
> > > - Explicit derivation of the ELBo loss in an appendix. As it is typically used, the purpose of an ELBo loss is to lower the KLD between an approximating function and the true posterior. Here, it is unclear what is playing the role of the "true posterior." The fact that $p_{\mathcal{T}}$ is the second argument to the KLD seems to indicate that $p_{\mathcal{T}}$ serves this function. However, the authors say that $p_{\mathcal{T}}$ is "learned", which seems to contradict this.
> > > - Finally, the role of the first term in equation 5 still needs to be explained. Typically, this would be the log probability of the true latent states $z$ under the approximating function. Instead, it is the expectation w.r.t. the approximating function of the true graph representation. I can intuitively see how this plays the same role, but there is a non-trivial step being made here that should be noted.
> > >
> > > The authors make a good attempt at a hard problem. As it is, a complex presentation makes it difficult to judge the soundness of their work completely. But for the most part, their design seems promising.
> > >
> > > # minor concerns
> > > > From Section 4: baselines: GATA (Adhikari et al., 2020) that learns a graph-structured representation to model the environment dynamics
> > > - This makes it sounds like GATA is a model-based system.

---

> > > > ### Author Response · Authors · 2021-11-30
> > > > **Thanks for your reply and response to your concerns**
> > > >
> > > >
> > > > Dear reviewer, we are greatly thankful for your reply. We hope the following response can solve your concerns.
> > > >
> > > > 1. *"Clarifying what the output of training is: Is it $p_{\mathcal{T}}$ or $q_{\epsilon}$? ... It seems to me that $p_{\mathcal{T}}$ is redundant, ..."*
> > > >
> > > > - ***Response:*** We agree that the relation between $p_{\mathcal{T}}$ and $q_{\epsilon}$ requires more explanation, and we will add them in the final version. $q_{\mathcal{T}}$ and $q_{\epsilon}$ are the conditional prior and the conditional approximate posterior in the ELBo (see more explanation in the next response). Both of them will be updated by ELBo during training, and the agent needs to use both of them to solve the text game. Figure 1 shows an example. We use $q_{\mathcal{T}}$ ***during planning*** (e.g., with MCTS) where the game environment is not available and we can ***not*** depend on the observation. We use $q_{\mathcal{T}}$ to predict the belief state. However, ***After*** the planner decides an action, we use the action to interact with the environment and obtain an observation. This observation is applied to update the belief with the posterior $q_{\epsilon}$. In this sense, model-based RL can significantly improve sample efficiency since the planner does not require samples from the environment. We will add these explanations in the revised version.
> > > >
> > > >
> > > > 2. *"Explicit derivation of the ELBo loss in an appendix... Here, it is unclear what is playing the role of the "true posterior."*
> > > >
> > > > - ***Response:*** We agree that derivation of the ELBo requires more discussion. In short, $q_{\mathcal{T}}$ and $q_{\epsilon}$ are the conditional ***prior*** and the conditional approximate ***posterior*** in the ELBo. The order of the prior and the posterior in the KLD term is consistent with that in the fundamental work for VAE [1] (check their equation 3). We agree this order is not intuitive and we will show how to derive such an order in the revised version.
> > > >
> > > > [1] Kingma, Diederik P., and Max Welling. "Auto-Encoding Variational Bayes." ICLR 2014.
> > > >
> > > > 3. *"Finally, the role of the first term in equation 5 still needs to be explained..."*
> > > >
> > > > - ***Response:*** We admit the terms in our ELBo (equation 5) are not very intuitive. Actually, the first term in equation 5 can be understood as a reconstruction loss in the VAE. You are right that using the log probability of the true latent states is very ideal, but the true latent state can not be observed (from the environment or from the model) and we can not use it to supervise the training. We instead use the graph since the latent states are supposed to embed the important information in the graph. We will add the explanation in the revised version.
> > > >
> > > > 4. *"This makes it sound like GATA is a model-based system."*
> > > >
> > > > - ***Response:*** GATA is a model-free agent based on Q-learning. They basically learn a graph representation for the states by following a Conditional Auto-Encoder structure, so their Q function depends on the learned states instead of the text. They didn't build any planner or transition functions. We will clarify this in the revised version.

---

### Official Review · Reviewer_W88S · 2021-11-02

**Correctness:** 3
**Technical Novelty And Significance:** 3
**Empirical Novelty And Significance:** 3
**Recommendation:** 5
**Confidence:** 3

**Main Review:**

Strength:

1. This paper studies the TextGames, a very interesting and exploratory direction, and sets up a new SOTA performance through the model-based RL methods. The success enriches the application of dynamic model learning and planning algorithms. It may encourage follow-up works along the direction of planning under high-level semantic information and inspire fields across visual language navigation and robotic manipulation.
2. Though there are existing works that train object-oriented dynamic models, applying them to text-domain requires non-trivial efforts. The paper demonstrates the importance of an object-oriented representation in text-model learning, which may help future research in learning better language models. If I am correct, the proposed method assembles various recent approaches, providing a very reasonable foundation for future researches.

Weakness

1. Though the paper does well in presenting the core idea of learning an object-oriented dynamic model, I met difficulties while parsing the architecture details. I believe that there is still a large room for improvement regarding paper writing. Here is my advice
    1. The overall structure is a little bit confusing. The paper seems to mix the problem setup with the proposed method. For example, the paper introduces the OO-MDP, graph representation, and model-based planning in Sec 2, and introduces the detailed transition model, reward model, and training methods, and additional modules in Sec 3. This causes the structure fragile.
    2. For example, the paper introduces the concept of learning dynamic model in the section defining OOMDP and mentions a learned graph updater in the section defining memory graph. In section 2.3, it uses Table 1, which contains a lot of undefined terms, before introducing the planning methods. As a result, after reading Sec 2, I do not know what the graph updater is and whether the latent z is the ground truth state or a learned latent state. In fact, until now I do not understand the meaning of the graph updater as I can not find any strict definition about it.
    3. The training objectives section contains too much content. The authors put two different training setups, the way to construct the dataset, and several important network modules (observation encoder, decoder, and so on) in the last small section. This violates the logical flow and makes the section less coherent. The consequence is that I have to read back and forth to figure out what the networks actually learn and even go back to Sec 2 to get the meaning of variables.
    4. While introducing the planning method and the transition model, the authors mention that "Given/define the states z". It is not proper, as z should be a learned latent variable and I think there is no ground truth information about z. In contrast, the authors should talk more about the observation, which is the real given input to the dynamic model. Currently, the observation encoder q_Epsilon is never formally defined and is missing in Figure 2.
    5. In sec 2.2, the paper defines G_t, but it forgets to define h_t and oplus operator, making the graph updater unclear.
    6. It forgets to mention the distribution that the model uses. Are they all Gaussians in formula (5) and (7)?
    7. Section 3.1(c) lacks explanation. More importantly, it does not explain how it generates the two attention matrices Bs. Are they generated by the BIDAF networks? What is the input to that network?
2. It seems that the model requires knowing the object set before training the model. Even in the self-supervised setting, the model needs to use a pre-determined object set. I worry if introducing too much prior will hurt the generalizability of the proposed method in new language tasks.

Minor comments:

1. I don't agree that extending OO-MDP with partial observability is a significant contribution.
2. In Figure 4, there are two OS-OOTD in the title.
3. Regarding the performance of the OS and SS setup, the paper mentions that OS's inferiors are due to the inaccurate reward prediction. I am not fully convinced as the adding extra supervision should not hurt the performance too much. Instead, I suspect that the reward module is too simple. Only a single MLP may not be enough. I suggest authors do more study about the reward module to figure out the reason behind the bad reward prediction.

**Summary Of The Paper:**

This work proposes Object-Oriented Text Dynamics for model-based RL in the TextWorld environment. OOTD ensembles existing methods such as ComplEx, R-GCN to build a memory graph, which contains object-level information about each object's attribute and its relations to others, from history observations and predicts states at the next time step. Therefore, the authors can combine the learned model with Dyn-Q or MCTS methods to select actions. If I am correct, under an object-supervised setting, it assumes the existence of a dataset that contains a ground truth memory graph. In contrast, the network has to discover graphs from predicting observation and rewards in the self-supervised setting. Experiments show that OOTD achieves superior performance than various baselines, including model-free RL and previously learned graph models.

**Summary Of The Review:**

I appreciate the author's effort in constructing such a complicated model to solve the challenging tasks, but the paper's writing has big flaws. Without adding enough details for a better evaluation, I do not recommend accepting this paper in its current form. I would be happy to increase my score if its writing can be improved.

---

> ### Author Response · Authors · 2021-11-14
> **Response to Reviewer W88S -- Part1**
>
> Dear Reviewer, we appreciate your constructive comments. We have revised the paper with the following clarification as per your feedback,
>
>  1. *"In section 2.3, it uses Table 1, which contains a lot of undefined terms, before introducing the planning methods."*
> - ***Response:*** The important components in Table 1 are defined in Section 2.2 and Section 2.3. $k$ denotes the $k^{th}$ object. We have a total of $K$ objects and thus a set of $K$ transition models and a set of $K$ encoders.
>
> 2. *"I do not know whether the latent $z$ is the ground truth state or a learned latent state."*
> - ***Response:*** $z_{k,t}$ is the learned latent state. $z_{k,t}$ denotes the state for the $k^{th}$ object. The object states are represented by latent values and thus we call them latent state. $z_{k,t}$ is not directly returned by the environment, but instead the output of the transition models and encoders.
>
> 3. *"I do not know what the graph updater is."*
> - ***Response:*** The graph updater predicts $h_{t}$ from $h_{t-1}$, as we mentioned in Section 2.3. This function is integrated into the OOTD model . To better clarify it, in Figure 2, if we continue the generation by inputting the predicted $z_{t}$ into the next graph encoder, we can get $h_{t}$. In this sense, the OOTD model can predict $h_{t}$ from $h_{t-1}$.
>
> 4. *"The training objectives section contains too much content. The authors put ... in the last small section. This violates the logical flow and makes the section less coherent."*
> - ***Response:*** To develop more reader-friendly writing, we have ***modified the structure*** of this section by emphasizing the important components. We agree that the training objectives consist of many components. Covering all the details of these components in a paper with strict page limits is challenging, so we refer to many pre-defined concepts and move a few things that are less important to appendices.
>
> 5. *"The authors mention that "Given/define the states z". It is not proper, as z should be a learned latent variable and I think there is no ground truth information about z. In contrast, the authors should talk more about the observation, which is the real given input to the dynamic model."*
> - ***Response:*** When we write that $z$ is given, we do not mean that it is observed, but simply that it is provided as input (i.e., given) to OOTD at the current time step after being computed by OOTD at the previous time step. $z_{t}$ indicates the predicted object states instead of values directly returned by the environment or labels from datasets. We added a clarification about this in the paper. In terms of observations, they are not available ***during*** planning since the planner will interact with learned dynamics models instead of the environment, so OOTD can not condition on observations. However, the belief of object states will be updated by the observation encoder ***after*** the planner determines an action (that we execute in the environment).
>
> 6. *"Currently, the observation encoder $q_\epsilon$ is never formally defined and is missing in Figure 2."*
> - ***Response:*** In fact, we have briefly defined $q_\epsilon$ after the objective (5) (Section 3.3). We agree this definition is not detailed enough and we have provided a very detailed definition in in appendix A.5. The purpose of Figure 2 is to illustrate the transition function (the conditional prior in the ELBo objective), so we omit $q_\epsilon$ (the approximate posterior in ELBo). We have visualized the structure of $q_\epsilon$ in Figure 6.
>
> 7. *"In sec 2.2, the paper defines $G_t$, but it forgets to define $h_t$ and oplus operator, making the graph updater unclear."*
> - ***Response:*** We admit the we used the symbol $\mathcal{G}$ to represent different things. In Section 2.2, $\mathcal{G}$ indicates the goal space, but in Section 2.3, we use it to denote a graph. We haved fixed this bug in the revised version. the purpose of Section 2.2 is to define OOPOMDP, whereas the graph and its updater is not part of it, so we separate their definitions into two sections.
>
> 8. *"It forgets to mention the distribution that the model uses. Are they all Gaussians in formula (5) and (7)?"*
> - ***Response:*** In Equation (3), Equation (8), Figure 2 and Figure 6, we mentioned that they are paramterized by Gaussian distributions. We followed common CVAE designs and modeled the distribution of $z$ with independent Gaussians. We have emphasized this in the revised version.

---

> > ### Author Response · Authors · 2021-11-14
> > **Response to Reviewer W88S -- Part2**
> >
> > 9. *"Section 3.1(c) lacks explanation. More importantly, it does not explain how it generates the two attention matrices Bs. Are they generated by the BIDAF networks? What is the input to that network?"*
> > - ***Response:*** Yes, the attention matrix $\boldsymbol{B}^{ae}$ is the output of BIDAF network. The inputs of this network are node representations $e_{t-1}$ and the action $a_{t-1}$, and that's why $\boldsymbol{B}^{ae}$ is able to compute the attentions between them. We have clarified this in the revise version.
> >
> > 10. *"It seems that the model requires knowing the object set before training the model... I worry if introducing too much prior will hurt the  generalizability of the proposed method in new language tasks."*
> > - ***Response:*** We admit using a pre-determined object set might limit the generalizability.  This assumption is consistent with prior work that also utilized knowledge of the set of objects [Adhikari et al., 2020] . Avoiding this assumption is a great direction for future work.
> >
> > [Adhikari et al., 2020] Ashutosh Adhikari et al.,. Learning dynamic belief graphs to generalize on text-based games. NeurIPS 2020.
> >
> > 11. *"In Figure 4, there are two OS-OOTD in the title."*
> > - ***Response:*** We have fixed it.
> >
> > 12. *" Regarding the performance of the OS and SS setup, the paper mentions that OS's inferiors are due to the inaccurate reward prediction. I am not fully convinced as the adding extra supervision should not hurt the performance too much. Instead, I suspect that the reward module is too simple. Only a single MLP may not be enough. I suggest authors do more study about the reward module to figure out the reason behind the bad reward prediction."*
> >
> > - ***Response:*** Our reward function is implemented by a transformer+mlp model (Section 3.2). This structure should be powerful enough for reward prediction. More importantly, both the OS and SS setup apply the same reward model for a fair comparison. The issue of reward model structure might not be an ideal explanation for the difference. In fact, a previous work (Lambert et al., 2020) has explored this issue and named it "objective-mismatch problem" in model-based RL. In summary, the RL agent is optimized to maximize rewards, whereas the OS-based dynamic model is optimized to predict object states. Compared to OS, the SS-based dynamic model, which is optimized to predict rewards, has a closer objective to that of the RL agent. We believe the mismatch between the objectives of dynamic learning and RL controlling provide a better explanation..
> >
> > [Lambert et al., 2020] Nathan O. Lambert, Brandon Amos, Omry Yadan, and Roberto Calandra. Objective mismatch in model-based reinforcement learning. Proceedings of the 2nd Annual Conference on Learning for Dynamics and Control, L4DC 2020.

---

> > > ### Comment · Reviewer_W88S · 2021-11-30
> > > **Response to Authors**
> > >
> > > Thanks for your update. I believe this is a good paper with a minor issue with the writing. Your comment has clarified many of my concerns but I am still worried about the overall structures and the missing details which may be put in an appendix. I will keep my score but I will not argue for a rejection.

---

### Official Review · Reviewer_DWsm · 2021-11-03

**Correctness:** 4
**Technical Novelty And Significance:** 2
**Empirical Novelty And Significance:** 3
**Recommendation:** 6
**Confidence:** 2

**Main Review:**

The motivation of the paper makes sense. The author starts with challenges in text-based games:
* text observation has variable length of sequence.
* text only partially describes the current state
* text could contain noisy information and it is challenging to identify salient terms.

General Significance: This is a difficult and interesting domain to study. The author proposes a reasonable extension based on OO POMDPs for text-based environment. The empirical results give strong evidence that this approach led to reasonable improvement in planning.

Novelty: For my best knowledge, I think the method is a new contribution to the field.

Technical Quality and Clarity: The paper is well-structured, with clear relationship between each section. The general ideas are clearly conveyed. I especially like Figure 1 and Figure 4. Figure 1 is illustrative and shows the high-level idea clearly. Figure 4 visualizes the object state embedding, which clearly shows that the OS-OOTD method was able to embed semantic similar objects (ingredients) in a near position.  But if the author can use more illustrative examples on explaining the technical details, it would be more readable for broader audience.

Typos:
* Figure 4: the title for the right-most image should be "SS-OOTD".

**Summary Of The Paper:**

This paper studies object-oriented text dynamics (OOTD) model and planning in text-based games. OOTD learns internal representation of object dynamics and uses transition layers to predict the belief of object states. Empirical results shows a performance boost compared to model-free baselines. Ablation studies are performed to explain the importance of OOTD components.

**Summary Of The Review:**

marginally accept.

---

> ### Author Response · Authors · 2021-11-14
> **Response to Reviewer DWsm**
>
> Dear Reviewer, we appreciate your constructive comments and provide our response to your concerns as follows.
>
> 1. *"if the author can use more illustrative examples on explaining the technical details, it would be more readable for broader audience."*
> - ***Response:*** We added the detailed implementation of the goal extractor, the object extractor and the observation encoder in Appendix A.3, A.4 and A.5.
>
> 2. *"Typos"*
> - ***Response:*** We have fixed them.

---

> ### Comment · Reviewer_DWsm · 2021-11-29
> **Acknowledgement of author response**
>
> Thanks for the response. I am keeping my ratings.

---

### Author Response · Authors · 2021-11-30
**A summary of our updates**

Dear Reviewers, Area Chairs, and Program Chairs,

We are greatly thankful for the insightful comments and suggestions, which are very helpful for us to further improve this work. We believe the reviewers generally agree that our submission is technically sound and the (novel) oriented-oriented idea is clearly conveyed. We admit some clarifications, explanations, and model details should be added. To clarify our modifications and prevent misunderstanding, we summarize our major updates in the following:

1. We have rewritten section 3.3 according to the suggestions and comments from our reviewers. In this section, we clarify the definition of our training objectives and the relation among the components in our objectives (especially about the role of our memory graph $h_{t}$ in learning).

2. We have added more details about the goal extractor, object extractor, and object extractor by introducing and visualizing their model structures in Appendix A.3, A.4, and A.5.

3. We have updated the symbols of goal and graph by using $\mathcal{G}$ and $\Omega$ to represent the goal space and the extracted graph at a time step (Section 2.2 and 2.3). Now, we can safely define the memory graph by aggregating previous extracted graph: $h_{t}=\Omega_{0}\oplus\Omega_{1} \oplus\dots\oplus\Omega_{t}$.

4. We have clarified the definitions of our object state $z$ by introducing that 1) $z$ is a learned latent state and 2) the relation between text states $s$ and object states $z$ (Section 2.2).

5. We have clarified the inputs and the outputs of our BIDAF networks and the role of BIDAF in our OOTD model (Section 3.1 (c)).

6. We have fixed the title in Figure 4.

---

### Decision · Program_Chairs · 2022-01-20

**Decision:**

Accept (Poster)

**Comment:**

This manuscript makes an interesting observation: there is no reason why planning-based methods like MDPs must be limited to physical or grounded environments. One can plan about more abstract textual domains. It adapts the standard methods from planning to such text domains in a fairly straightforward way. The fact that concepts from MDPs map to these problems directly is an asset: ideas could flow between these domains in the long term. While the original submission was lacking clarity and significant technical details, the authors engaged with the reviewers and resolved lingering concerns. Reviewers are unanimous that this a strong contribution.